# Applying Visual Storytelling in Food Marketing: The Effect of Graphic Storytelling on Narrative Transportation and Purchase Intention

**DOI:** 10.3390/foods14152572

**Published:** 2025-07-23

**Authors:** Lingnuo Wang, Xin Fang, Ying Xiao, Yangyue Li, Yulin Sun, Lei Zheng, Charles Spence

**Affiliations:** 1School of Business, Macau University of Science and Technology, Taipa, Macau; wanglingnuo0101@163.com (L.W.); xfang@must.edu.mo (X.F.); 1240012321@student.must.edu.mo (Y.L.); 1210009703@student.must.edu.mo (Y.S.); 2Faculty of Medicine, Macau University of Science and Technology, Taipa, Macau; yxiao@must.edu.mo; 3Crossmodal Research Laboratory, Department of Experimental Psychology, University of Oxford, Oxford OX1 3UD, UK; charles.spence@psy.ox.ac.uk

**Keywords:** graphic storytelling, food consumption, narrative transportation, cognitive fluency

## Abstract

In today’s market, consumers appear to be less interested in promotional strategies, particularly those that rely on text-based advertisements. Graphic storytelling can be seen as providing a more engaging visual approach to attract audiences and is increasingly being used by marketers and food packaging designers. However, the questions of whether and how graphic storytelling influences consumers’ purchase intentions remain underexplored. Based on the Transportation–Imagery Model, two experimental studies were conducted to examine the effect of graphic storytelling on narrative transportation and food purchase intention, and to explore its underlying mechanism from the perspective of cognitive fluency. The results demonstrated the positive effect of graphic storytelling on narrative transportation (Studies 1 and 2), as well as a significant impact on food purchase intention (Study 2). Furthermore, cognitive fluency was identified as a critical factor impacting narrative transportation, facilitated by graphic storytelling (Studies 1 and 2). This study extends the Transportation–Imagery Model by positioning cognitive fluency as an important antecedent of narrative transportation. Practically, the suggestion would be for restaurants and food firms to optimize their advertising by displaying cooking processes, particularly for part-prepared foods.

## 1. Introduction

In today’s competitive marketplace, brands are increasingly turning to engaging visual strategies to capture consumer attention, as consumers often rely on their visual systems to select products [1,2]. One such strategy is the use of graphic storytelling on product packaging, particularly in the food industry, where it is increasingly used to illustrate food preparation steps or to provide cooking instructions for prepackaged meals. Graphic storytelling, also referred to as visual storytelling, merges the strengths of both images and text, offering a more dynamic and effective means of communication [3,4,5]. The existing research indicates that graphic storytelling demands less cognitive effort to process information than textual descriptions, while conveying more information than a single static image [6,7]. This dual advantage enhances comprehension and engagement [8,9]. Thus, graphic storytelling has emerged as an important tool in the context of food marketing. Surprisingly, limited research has explored the impact of graphic storytelling on food purchase intention and its underlying psychological mechanisms. To address this gap, we conducted two scenario-based experiments to investigate how cognitive fluency mediates the relationship between exposure to graphic storytelling, narrative transportation, and food purchase intention. Specifically, Study 1 tested these effects in the context of part-prepared food packaging, while Study 2 examined the effects in a restaurant advertising context to evaluate the generalizability of the proposed mechanism. The results consistently showed that graphic storytelling enhances cognitive fluency, which in turn increases narrative transportation, ultimately leading to greater purchase intention. Thus, this research extends the Transportation–Imagery Model by identifying cognitive fluency as a key antecedent of narrative transportation and deepens the understanding of how graphic storytelling influences consumer decision-making.

## 2. Literature Review

### 2.1. Graphic Storytelling, Narrative Transportation, and Purchase Intention

Graphic storytelling is a form of communication that conveys stories through a single picture or a series of images arranged in a logical sequence [3,4,10,11]. Although multi-panel comics are often regarded as more effective for storytelling [4,10], a single image can also convey a narrative through visual cues, embodying the adage “a picture is worth a thousand words [11]". This research adopts a text-free, character-absent visual sequence, designed with causal and temporal coherence. This organizational structure enables consumers to perceive a coherent beginning, middle, and end. To fully comprehend the narrative, consumers must actively integrate visual elements across panels [3,10]. This process facilitates causal inference, allowing viewers to understand how individual events are connected and how one action leads to another [12]. Thus, graphic storytelling is regarded as a powerful communication tool that enhances both comprehension and consumer engagement [4,5,13].

According to the Transportation–Imagery Model [14,15], narrative transportation is composed of three elements: cognitive attention, mental imagery, and emotional engagement. When transportation occurs, consumers become mentally disengaged from their immediate surroundings and emotionally involved in the narrative [15,16,17]. According to past research, visual elements used in graphic storytelling have been shown to enhance narrative engagement while reducing cognitive resistance, thereby strengthening the persuasive impact of the message [18,19]. Thus, by using images to convey information, graphic storytelling decreases consumers’ dependence on their own imagery ability to interpret the story, particularly because these narratives are presented through verbal or textual descriptions [13]. Graphic storytelling highlights action–outcome links [10], thus allowing consumers to mentally rehearse the sequence of events (e.g., “If action X occurs, then outcome Y will result”) [20], which ultimately enhances narrative transportation [15]. As a result, graphic storytelling is thought to increase purchase intention by decreasing cognitive resistance and minimizing negative emotional responses [18,19]. Thus, we hypothesize that graphic storytelling can enhance narrative transportation (H1*a*), which further increases purchase intention (H1*b*).

### 2.2. Cognitive Fluency

Cognitive fluency refers to consumers’ metacognitive experience of how easily they process information [21,22,23]. Specifically, when consumers encounter information that is easier to process, they are more likely to evaluate it favorably and perceive it as truthful and credible, especially in the contexts of advertising and online shopping [24,25]. Research has shown that cognitive fluency significantly affects various consumer outcomes, including purchase intention and product liking [26], positive attitudes toward advertisements and brands [27], and satisfaction with mobile online stores [23].

Graphic storytelling is characterized by its inherent coherence and its ability to convey information through vivid images, which are believed to be more easily and fluently understood [8,9]. Specifically, consumers experience higher cognitive fluency when information is structured in a logically connected and causally coherent sequence [9,28]. For example, structured word triads with consistent flow are processed more fluently, which results in a brief and subtle positive affect [29]. Compared to verbal narratives, graphic storytelling has been found to facilitate understanding more effectively, particularly when the message is complex or abstract [13]. This is because it leverages vivid imagery to present specific, concrete details enriched with perceptual, semantic, and contextual cues, thus enhancing the ease of comprehension [5,8,27]. For instance, the Harry Potter film visually depicts the wand, allowing viewers to perceive its appearance directly, whereas the novel describes it more abstractly as “eleven inches, nice and supple.” Such specific and concrete information that is rich in perceptual, semantic, and contextual details is thought to enhance cognitive fluency [5,8,27]. By integrating details into a coherent visual narrative, graphic storytelling may reduce ambiguity and minimize the cognitive effort required for consumers to construct mental representations on their own.

Cognitive fluency enhances the formation of mental imagery and encourages active engagement, both of which are crucial for enhancing narrative transportation [21,30,31,32]. Specifically, when consumers are presented with fragmented or cognitively demanding information, they have to allocate additional mental resources to comprehend it. In contrast, fluent processing reduces cognitive load, allowing consumers to allocate more mental resources to mental simulation [6,21]. This mental imagery plays a crucial role in enhancing narrative transportation, providing consumers with an immersive experience [15]. Indeed, narrative transportation itself is closely related to fluency: it occurs more readily when information is processed smoothly and when consumers construct vivid internal representations of the narrative [6,21,33]. Therefore, cognitive fluency acts as a key driver in narrative transportation [21,34], which, in turn, affects consumer decision-making. Taken together, these findings suggest that when graphic storytelling improves cognitive fluency, it not only facilitates understanding but also deepens consumers’ emotional involvement with the narrative. We hypothesize that cognitive fluency mediates the relationship between graphic storytelling and narrative transportation (H2*a*), which subsequently leads to an increased purchase intention (H2*b*).

### 2.3. The Present Study

Graphic storytelling is increasingly used in advertisements and food packaging as an effective means of visually communicating information [11,35]. This research consisted of two experimental studies that were designed to examine the impact of graphic storytelling on consumers’ narrative transportation and purchase intention in the context of food products. Additionally, the underlying psychological mechanism was studied from the perspective of cognitive fluency (Figure 1). Specifically, Study 1 focused on part-prepared food products and investigated how graphic storytelling impacted narrative transportation, with cognitive fluency examined as a potential mediating variable. Study 2 extended these findings to a restaurant advertising context, further exploring the effects of graphic storytelling on narrative transportation and purchase intention, while again testing the mediating role of cognitive fluency in these relationships.

## 3. Study 1: Examining the Effect of Graphic Storytelling on Narrative Transportation and the Mediating Role of Cognitive Fluency

Study 1 was designed to examine the effect of graphic storytelling in advertising on narrative transportation and to investigate the mediating role of cognitive fluency. Using part-prepared food, we hypothesized that presenting graphic storytelling (vs. the control) in advertisements would lead to greater narrative transportation, and that any such effect would be mediated by cognitive fluency.

### 3.1. Participants and Procedure

A total of 200 participants were recruited from the Credamo platform (www.credamo.com) and provided informed consent. Only those who read and accepted the online consent form were allowed to continue, while those who declined were redirected to a closing page. Twenty-eight participants were excluded from the analysis due to failing an attention check question. The final sample consisted of 172 participants (Mage = 31.63 ± 8.56 years), including 117 women (Mage = 31.32 ± 7.68 years) and 55 men (Mage = 32.31 ± 10.21 years). The participants were randomly assigned to either the graphic storytelling group (viewing an ad depicting the process of dish preparation) or the control group (viewing an ad showing the ingredients in the dish; Figure 2). Specifically, they were informed that “Over the weekend, you see a box of part-prepared sauerkraut fish in the supermarket priced at 38 Yuan (RMB).” They were then presented with one of two packaging descriptions, depending on their assigned condition: [Graphic storytelling group: the packaging shows the production process of premade sauerkraut fish, including add water to boil, add soup base, fish fillet, and spices to stir-fry/Control group: the packaging lists the dish’s ingredients, including fish fillet, sauerkraut, sharp red pepper, and peppercorns.] The participants viewed the stimulus materials and completed the subsequent measures without any time constraints. In this study, the participants completed measures, including narrative transportation, cognitive fluency, and a manipulation check (see details in Appendix A), as well as demographic information (e.g., sex, age, and educational level). The entire study required approximately 3–4 min per participant. Upon completion, the participants received ¥1 RMB (about USD 0.14) as a thank-you through Credamo.

### 3.2. Measures

Narrative transportation was adopted from the questionnaire developed by Green and Brock [36]. Four items were included in this study, such as “While viewing the packaging, I could easily picture the event taking place.” The participants responded on a 5-point scale (1 = strongly disagree, 5 = strongly agree). The Cronbach’s alpha for this sample was 0.86.

Cognitive fluency was measured using a question adapted from Graf, Mayer, and Landwehr [22]. The participants were asked: “The process of understanding the packaging was…” with responses anchored on the following scales: difficult to easy, unclear to clear, disfluent to fluent, incomprehensible to comprehensible. The Cronbach’s alpha for this sample was 0.87.

### 3.3. Manipulation Check

The two-item manipulation check (“The packaging showed the food preparation process in sequence” and “The packaging allowed me to see the food preparation process in sequence”) was rated on a 5-point scale (1 = strongly disagree, 5 = strongly agree; *r* = 0.89). The results demonstrated that the manipulation was successful (as in the graphic storytelling group: M_graphic storytelling_ = 4.44 ± 0.74; M_control_ = 3.36 ± 1.36; F[1,170] = 41.33, *p* < 0.001,
$\eta_{p}^{2}$ = 0.20).

### 3.4. Results

The participants in the graphic storytelling group scored higher in both narrative transportation (M = 5.82 ± 0.79; F[1,170] = 18.32, *p* < 0.001,
$\eta_{p}^{2}$ = 0.10) and cognitive fluency (M = 4.51 ± 0.38; F[1,170] = 17.07, *p* < 0.001,
$\eta_{p}^{2}$ = 0.09) compared to those in the control group, respectively (narrative transportation: M = 5.13 ± 1.26; cognitive fluency: M = 4.08 ± 0.87).

To examine the mediating role of cognitive fluency, a mediation analysis was conducted using PROCESS Model 4 with 5000 bootstrap resamples. The results indicated that graphic storytelling (vs. the control) significantly predicted cognitive fluency (*β* = 0.60, SE = 0.10, *p* < 0.001), which further predicted narrative transportation (*β* = 0.66, SE = 0.09, *p* < 0.001), confirming the mediating role of cognitive fluency (Effect size = 0.40, SE = 0.08, 95% CI [0.24, 0.55]; see Figure 3).

### 3.5. Discussion

The results of Study 1 demonstrate that graphic storytelling (e.g., product preparation imagery) enhances both cognitive fluency and narrative transportation when consumers process ad information for a part-prepared food product. Since part-prepared foods require consumers to be involved in the preparation process, the latter may naturally experience greater narrative transportation. To assess whether this observed effect persists when the food is prepared by someone else, and to evaluate the generalizability of the findings beyond packaged goods, Study 2 utilized a restaurant advertising context. Furthermore, the effect of graphic storytelling on narrative transportation and its subsequent impact on purchase intention was re-examined.

## 4. Study 2: The Effect of Graphic Storytelling on Narrative Transportation and Purchase Intention: The Mediating Role of Cognitive Fluency

In Study 2, it was hypothesized that presenting graphic storytelling (vs. the control) would lead to higher levels of cognitive fluency and stronger narrative transportation, which would further enhance purchase intention.

### 4.1. Participants and Procedure

This study recruited 120 participants from Credamo (www.credamo.com). Only those who read and accepted the online consent form could proceed, while others were redirected to the closing page. A total of 17 participants (14.17%) were excluded due to failing an attention check question (see Appendix A for details), leaving 103 participants for final analysis (Mage = 26.77 ± 6.38 years; 36.89% women). Among them were 38 women (Mage = 26.53 ± 6.50 years) and 65 men (Mage = 26.91 ± 6.35 years). All participants were informed that they would be completing a survey about soup dumplings. Then, the participants provided their informed consent and were randomly assigned to either the graphic storytelling group (viewing an ad depicting the process of dish preparation) or the control group (viewing an ad showing the ingredients of the dish; Figure 4). Specifically, the participants were informed that “You are a faculty member working in the central business district. At noon, you see an advertisement flyer for a restaurant near your workplace promoting its dish—*Tang Shui Jiao* (soup dumplings). The flyer presents [Graphic storytelling Group: the preparation process, including mixing the meat filling, rolling the dumpling wrappers, and wrapping the filling]/[Control Group: the dish ingredients, including fresh vegetables, fresh pork, and dumpling wrappers]. The *Tang Shui Jiao* is priced at 24 Yuan (RMB) per serving.” The participants reviewed the visual materials and completed the measured tasks without any time constraints (taking approximately 3–4 min). The measures included purchase intention, narrative transportation, cognitive fluency, and a manipulation check (see details in Appendix A), as well as demographic information such as sex, age, and educational level. After completing the study, the participants received ¥1 RMB (approximately USD 0.14) through the survey platform.

### 4.2. Measures

The purchase intention was measured using two items (“You are about to have lunch, how willing are you to consume the food?”, “You are about to have lunch, would you be willing to consume the food?”). The participants responded on a 5-point scale (1 = Not at all, 5 = Very Much). The Cronbach’s alpha was 0.86 in this sample.

The measurements for narrative transportation and cognitive fluency were identical to those used in Study 1. The Cronbach’s alpha was 0.86 for narrative transportation and 0.79 for cognitive fluency, respectively.

### 4.3. Manipulation Check

The manipulation check measure (*r* = 0.94) was also the same as in Study 1. The results showed that the manipulation was successful (M_graphic storytelling_ = 4.61 ± 0.48; M_control_ = 2.37 ± 1.22; F[1,101] = 155.25, *p* < 0.001,
$\eta_{p}^{2}$ = 0.61).

### 4.4. Results

The participants in the graphic storytelling group scored higher on both narrative transportation (M = 4.31 ± 0.57; F[1,101] = 19.03, *p* < 0.001,
$\eta_{p}^{2}$ = 0.16) and cognitive fluency (M = 4.55 ± 0.42; F[1,101] = 8.91, *p* = 0.004,
$\eta_{p}^{2}$ = 0.08) as compared to those in the control group (narrative transportation: M = 3.71 ± 0.83; cognitive fluency: M = 4.23 ± 0.65). Moreover, the participants exhibited a stronger purchase intention in the graphic storytelling group (M = 4.30 ± 0.68; F[1,101] = 10.61, *p* = 0.002,
$\eta_{p}^{2}$ = 0.10) as compared to those in the control group (M = 3.77 ± 0.97).

In addition, a sequential mediation analysis was conducted using PROCESS Model 6 with 5000 bootstrap resamples [37]. The results showed that graphic storytelling (vs. the control) significantly predicted cognitive fluency (*β* = 0.57, SE = 0.11, *p* = 0.004), which further enhanced narrative transportation (*β* = 0.62, SE = 0.10, *p* < 0.001), ultimately increasing consumers’ purchase intention (*β* = 0.44, SE = 0.13, *p* < 0.001). The mediation analysis confirmed a significant sequential mediating effect of cognitive fluency and narrative transportation on the relationship between graphic storytelling and purchase intention (Effect size = 0.16, SE = 0.07, 95% CI [0.04, 0.29], Figure 5). However, when the order of the mediators was reversed (i.e., path from narrative transportation to cognitive fluency), the indirect effect was nonsignificant (effect size = 0.09, SE = 0.06, 95% CI [−0.06, 0.24]).

### 4.5. Discussion

Study 2 revealed that graphic storytelling (e.g., product preparation imagery) enhanced purchase intention through cognitive fluency and narrative transportation when consumers process packaging information concerning on-site meals. Notably, the mediation results revealed that graphic storytelling enhanced cognitive fluency, which in turn led to increased narrative transportation. Intriguingly, reversing this sequential mediating path (i.e., from narrative transportation to cognitive fluency) yielded nonsignificant results.

## 5. General Discussion

Using two experimental studies, this research demonstrated that graphic storytelling enhances narrative transportation through the mediating effect of cognitive fluency (Studies 1 and 2), which leads to increased purchase intention (Study 2).

### 5.1. Findings

These findings suggest that graphic storytelling significantly enhances narrative transportation, which subsequently increases purchase intention. According to prior research, effective narrative transportation typically requires three key elements: identifiable characters (e.g., protagonists), verisimilitude (the perception that the events could plausibly occur), and a coherent plot (a sequence of events unfolding over time) [15]. However, this research extends this literature by showing that graphic storytelling enhances narrative transportation in the context of food marketing, even in the absence of clearly identifiable characters. One possible explanation is that consumers implicitly infer the presence of a character (e.g., a chef, cook, or themselves) responsible for the preparation of the food. This inferred agency is considered to be sufficient to trigger narrative engagement. Chen et al. (2019) have suggested that listening to music fosters a sense of involvement in the events it depicts [38]. Although processing auditory and visual media involves distinct sensory channels, both cues are considered to trigger mental simulation and engage consumers [39]. For example, music leads consumers to detach from everyday concerns and evokes their cognitive and affective engagement within the musical atmosphere, which enhances their sense of immersion [40]. Another explanation is that verisimilitude and plot coherence may play a more dominant role in enabling narrative transportation than the presence of characters. Verisimilitude enables consumers to perceive depicted actions as events that either have occurred or are likely to occur [15]. It mitigates skepticism and leads consumers to perceive it as narratively plausible. In line with this, coherent plots are also regarded as essential for enabling consumers to engage in mental simulation [20]. For example, a visual sequence showing how a landscape changes with the seasons can effectively convey the passage of time, thereby creating a coherent narrative that draws viewers into the story. Aligning with that, prior research has demonstrated that a step-by-step process, such as pairing a Bluetooth speaker with an iPad, enhances narrative transportation by fostering engagement, clarity, and emotional connection [20]. This is because the step-by-step demonstrations provided by the store salesman reduce cognitive overload, promote a sense of flow, and evoke emotional resonance, all of which are essential for immersing participants in the process. Importantly, narrative transportation has been shown to evoke strong emotional responses and reduce critical thinking, both of which increase consumers’ purchase intention [7,15,32,41]. Our results support this view, highlighting the persuasive power of visual storytelling through graphic storytelling in the context of advertising.

Moreover, conveying information through graphic storytelling enhances cognitive fluency by providing inherent coherence and vivid visual presentation [8,9], both of which are essential for fostering effortless information processing. Prior research on narrative persuasion has suggested that engaging with narrative structures demands cognitive resources [6]. This contradiction may be explained by differences in communication media. Specifically, text-based narratives rely on readers’ imaginative capacity to construct mental imagery [13], which makes the comprehension process more effortful and cognitively demanding [6]. In contrast, graphic storytelling presents immediate visual depictions of narrative content [10], fostering more intuitive comprehension [27]. As a result, graphic storytelling delivers coherent narrative content while minimizing the cognitive effort required for information processing. Our findings suggest that both the coherence and the visual presentation inherent in graphic storytelling are essential for those wanting to foster cognitive fluency. Despite this, many existing studies on narrative advertising have focused on text-based formats when examining narrative effects [6,42,43]. Future research should further explore how different media formats, particularly those that integrate both visual and narrative elements, affect consumers’ cognitive processing and persuasive outcomes.

Notably, this research extends the Transportation–Imagery Model by identifying cognitive fluency as a mediator between graphic storytelling and narrative transportation. While [21] has proposed that cognitive fluency is an important antecedent of narrative transportation [21], our research provides direct empirical evidence demonstrating the effect of cognitive fluency on narrative transportation. Specifically, fluent processing leads consumers to allocate more cognitive resources toward imaginative stimulation, thereby facilitating mental imagery and participatory engagement [6,21], both of which are core components of the transportation experience [21,30,31,32]. In other words, the more easily a narrative is processed and visualized, the more deeply consumers become mentally immersed in it [7,15]. Conversely, when consumers perceive the content as disfluent, they are less likely to experience a sense of narrative engagement [44]. In essence, this research reveals a previously underexplored cognitive mechanism: the more fluently a narrative is processed and visualized, the more deeply consumers become mentally immersed. These findings underscore the importance of designing marketing narratives that not only possess a coherent structure but are also visually and cognitively accessible, particularly through formats such as graphic storytelling.

### 5.2. Managerial Implications

This study offers useful suggestions for researchers and professionals in the food industry on how to use graphic storytelling to increase consumer engagement and purchase intention. Graphic storytelling shows strong potential in the fast-growing market of part-prepared meals. While these products are known for being convenient, they are often seen as less natural or of lower quality [45]. However, packaging that shows step-by-step images of the cooking process (e.g., boiling soup or stir-frying ingredients) helps consumers feel more involved in preparing the meal. Even for part-prepared food products requiring only minimal preparation, such as microwave heating, explicitly detailing each procedural step (e.g., opening the packaging, placing the product in the microwave, and heating on high for one minute) can enhance consumers’ sense of involvement in the preparation process. This type of visual storytelling helps consumers feel more connected to the food [7,46], which may reduce their concerns about the food being too industrial or artificial. Thus, using graphic storytelling in packaging and advertising can play an important role in increasing consumer trust, acceptance, and ultimately, sales of part-prepared meals.

In addition, this research highlights the importance of cognitive fluency as a key factor that drives narrative transportation, which further increases purchase intention. To make graphic storytelling more effective, it is important to ensure that the visual sequence is logical, easy to follow, and clearly presented [28]. Specifically, to effectively illustrate preparation steps, images should feature high figure–ground contrast, clean and uncluttered layouts, and a well-defined visual hierarchy, complemented by universally recognized icons. These design elements enhance cognitive fluency, thereby facilitating easier processing of the content and increasing consumer engagement [23,47]. For example, employing numbered, step-by-step visuals can make instructions more intuitive and accessible, such as “Step 1: Chop vegetables” with a knife icon, “Step 2: Stir-fry ingredients” with a wok icon, and “Step 3: Add sauce” with a pouring icon. When information is presented in a cognitively fluent manner, consumers are more likely to become immersed in the narrative, positively influencing their attitudes and purchase decisions.

### 5.3. Limitations and Future Directions

There are several limitations. First, our research focuses mainly on static formats of graphic storytelling and does not explore video-based narratives. With the increasing use of digital platforms, such as TikTok and Instagram, short videos and other dynamic storytelling formats may have an even stronger impact on consumer behavior. Compared to single sensory input, multisensory interactions have a stronger effect on consumers’ perception of food [48,49]. Future studies should investigate how dynamic narratives influence consumers’ decision-making processes. For instance, research has demonstrated that, compared to rock music, classical music significantly enhances consumers’ preferences for healthy foods [50]. Building on this, future research could explore the interaction effects between different types of music within video-based storytelling and various food categories on consumers’ purchase intention. Moreover, future research could also explore the potential role of identifiable characters in narrative transportation by manipulating the presence or absence of a character in the video-based graphic storytelling. Second, our study relies on self-reported data, and no sample size determination was conducted. Both Study 1 and Study 2 used relatively small sample sizes. To strengthen the validity of these findings, future research should replicate the study in real-world settings, such as restaurant environments or supermarkets, to further examine the effect of graphic storytelling on purchase intention, and preregistration is recommended for future research. Additionally, utilizing objective measures, such as eye-tracking technology to monitor consumers’ attention to graphic elements, could offer more direct and precise evidence of their preferences. Third, this research did not explore boundary conditions under which graphic storytelling is more effective. Other potential factors, such as personality traits [51,52], may also contribute to explaining the effect of graphic storytelling on purchase intention. For instance, promotion-focused consumers may exhibit stronger purchase intention for foods presented with fancy and colorful graphic storytelling, whereas simple and minimalist graphic storytelling may be more appealing to prevention-focused consumers [51]. Future research should investigate these additional moderating effects to provide a more comprehensive understanding of the application of graphic storytelling in food marketing.

## 6. Conclusions

Through two experimental studies, this research demonstrated that consumers exposed to graphic storytelling were more likely to experience cognitive fluency and narrative transportation, which subsequently led to greater purchase intentions. This effect was consistently observed across both part-prepared food packaging and on-site restaurant advertising contexts.

## Figures and Tables

**Figure 1 foods-14-02572-f001:**
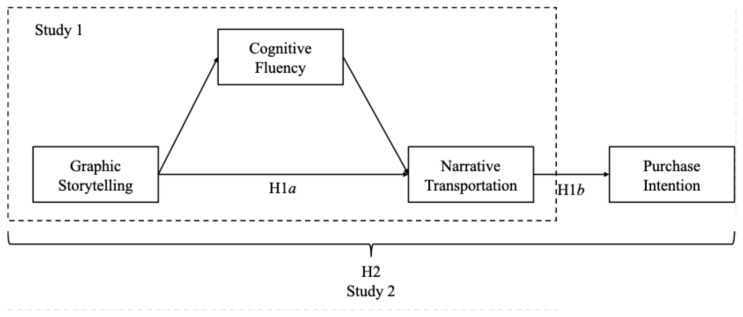
Conceptual model.

**Figure 2 foods-14-02572-f002:**
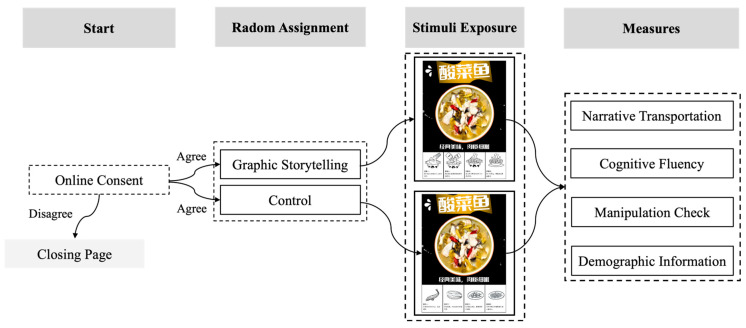
Research procedure for Study 1. Note. The Chinese text in the figure means "sauerkraut fish".

**Figure 3 foods-14-02572-f003:**
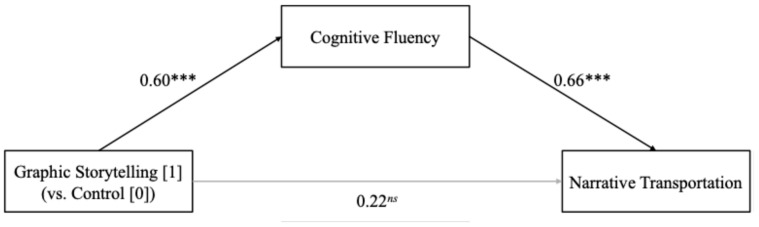
The mediating effect of cognitive fluency between graphic storytelling and narrative transportation. Note. *** *p* < 0.001, ns = non-significant.

**Figure 4 foods-14-02572-f004:**
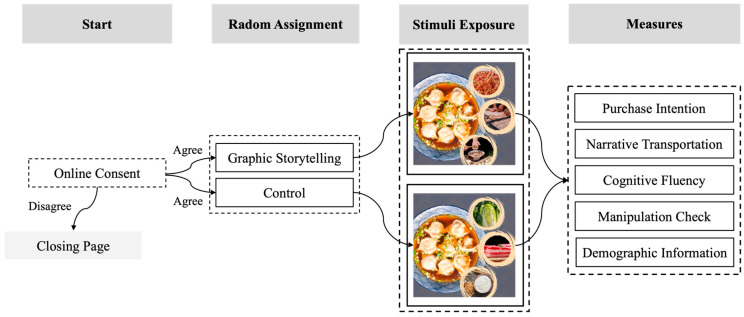
Research procedure for Study 2.

**Figure 5 foods-14-02572-f005:**
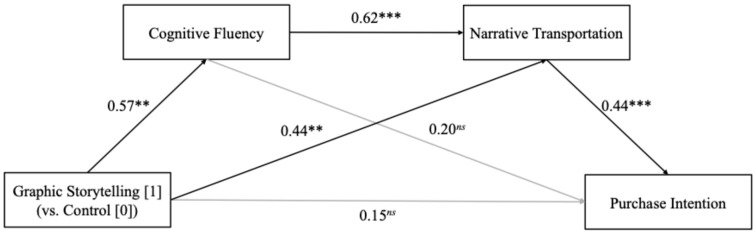
The mediating effect of cognitive fluency and narrative transportation on the relationship between graphic storytelling (vs. control) and purchase intention. Note. *** *p* < 0.001, ** *p* < 0.01, ns = nonsignificant.

## Data Availability

The original contributions presented in the study are included in the article, further inquiries can be directed to the corresponding author.

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
