# Peer review of "Applying Visual Storytelling in Food Marketing: The Effect of Graphic Storytelling on Narrative Transportation and Purchase Intention"

_foods, 2025, doi:10.3390/foods14152572_

Round 1

Reviewer 1 Report

Comments and Suggestions for Authors

Review: Applying Visual Storytelling in Food Packaging: The Effect of Graphic Storytelling on Narrative Transportation and Purchase Intention

This paper investigates the impact of graphic storytelling on food packaging on consumer narrative transportation and purchase intention, with a focus on cognitive fluency as a mediating factor. The research utilizes two studies to examine these relationships in different food contexts (part-prepared meals and restaurant advertising). The topic is highly relevant to current marketing trends, particularly the increasing use of visual strategies to engage consumers in the food industry. The authors have identified a gap in the existing literature regarding the effects and mechanisms of graphic storytelling in this domain. Overall, the paper presents interesting findings that contribute to the understanding of visual storytelling in food marketing. However, four areas could be improved to enhance the clarity, rigor, and impact of the research:

  1. Lack of Detailed Methodological Information:While the paper outlines the general procedure for both studies, a more comprehensive methodology section would significantly improve its transparency and replicability. The "Participants and Procedure" sections for both Study 1 and Study 2 are quite brief. They mention participant recruitment from Credamo and general assignments, but lack crucial details regarding the experimental design, specific instructions given to participants (beyond the general scenario), and the exact flow of the experiment. For instance, how long were participants exposed to the ads? The authors should expand the methodology sections to include a clearer description of the experimental design, including the time allocated for viewing stimuli and completing questionnaires and any debriefing procedures.
  2. Discussion of Absence of Clearly Identifiable Characters: The paper makes an interesting point about graphic storytelling enhancing narrative transportation even without identifiable characters, but this point could be expanded and potentially supported with more theoretical depth.The discussion states, "this research extends this literature by showing that graphic storytelling can enhance narrative transportation in the context of food marketing, even in the absence of clearly identifiable characters." (Lines 232-234) While explanations like implicit inference of a character (chef/cook/themselves) and the dominance of verisimilitude/plot coherence are provided, these are speculative. The authors should consider citing additional literature that discusses narrative transportation in abstract or process-oriented contexts, if available, to strengthen the theoretical backing for this observation. If possible, suggest avenues for future research (beyond the limitations section) that could empirically test these proposed explanations, e.g., by manipulating the presence or absence of a character in the graphic storytelling to see if the effects on narrative transportation differ.
  3. Managerial Implications - Specificity: The managerial implications are generally good but could be more concrete in certain suggestions.The suggestion to optimize advertising by "displaying cooking processes, particularly for semi-finished foods" isa good one. However, the advice for ensuring visuals have "high figure-ground contrast, simple layouts, and visual clarity" is somewhat generic. The authors should provide more specific examples or actionable advice for designers. For instance, rather than just "simple layouts," perhaps suggest "clean, uncluttered layouts with clear visual hierarchy," or "using universally understood icons for preparation steps." Illustrating with concrete examples of successful graphic storytelling in existing food packaging could also strengthen this section.
  1. Limitations and Future Research - Depth:The limitations are identified, but some could be expanded to suggest more innovative or impactful future research directions.The first limitation mentions static vs. video formats and briefly notes that dynamic formats might have a stronger impact due to sound, motion, and music. The suggestion to investigate "how dynamic narratives influence consumers' decision-making processes" is quite broad. For dynamic narratives, specify what aspects of sound, motion, or music could be investigated (e.g., the role of tempo in conveying preparation speed, specific musical cues for different food types). For the self-reported data limitation, beyond "real-world settings," consider suggesting specific physiological measures (e.g., eye-tracking to confirm attention to graphic elements) or behavioral measures (e.g., actual purchase data in a controlled store environment) to complement self-report. For the "other potential factors" limitation (e.g., food taste and personality), suggest how these could be integrated into the current model, perhaps as moderators or additional mediators, to build a more holistic framework. For example, how might perceived food taste interact with graphic storytelling to influence purchase intention?

Author Response

This paper investigates the impact of graphic storytelling on food packaging on consumer narrative transportation and purchase intention, with a focus on cognitive fluency as a mediating factor. The research utilizes two studies to examine these relationships in different food contexts (part-prepared meals and restaurant advertising). The topic is highly relevant to current marketing trends, particularly the increasing use of visual strategies to engage consumers in the food industry. The authors have identified a gap in the existing literature regarding the effects and mechanisms of graphic storytelling in this domain. Overall, the paper presents interesting findings that contribute to the understanding of visual storytelling in food marketing. However, four areas could be improved to enhance the clarity, rigor, and impact of the research:

Comments 1: Lack of Detailed Methodological Information:While the paper outlines the general procedure for both studies, a more comprehensive methodology section would significantly improve its transparency and replicability. The "Participants and Procedure" sections for both Study 1 and Study 2 are quite brief. They mention participant recruitment from Credamo and general assignments, but lack crucial details regarding the experimental design, specific instructions given to participants (beyond the general scenario), and the exact flow of the experiment. For instance, how long were participants exposed to the ads? The authors should expand the methodology sections to include a clearer description of the experimental design, including the time allocated for viewing stimuli and completing questionnaires and any debriefing procedures.

Response 1: We have added detailed information about the research protocol. For each study, we have outlined the experimental protocol to enhance clarity and understanding. Additionally, the exposure time and stimuli allocation have been included in the manuscript. We thank the reviewer for these suggestions (please see Pages 4-5, Lines 144-177; Pages 6-7, Lines 211-242).

Comments 2: Discussion of Absence of Clearly Identifiable Characters: The paper makes an interesting point about graphic storytelling enhancing narrative transportation even without identifiable characters, but this point could be expanded and potentially supported with more theoretical depth.The discussion states, "this research extends this literature by showing that graphic storytelling can enhance narrative transportation in the context of food marketing, even in the absence of clearly identifiable characters." (Lines 232-234) While explanations like implicit inference of a character (chef/cook/themselves) and the dominance of verisimilitude/plot coherence are provided, these are speculative. The authors should consider citing additional literature that discusses narrative transportation in abstract or process-oriented contexts, if available, to strengthen the theoretical backing for this observation. If possible, suggest avenues for future research (beyond the limitations section) that could empirically test these proposed explanations, e.g., by manipulating the presence or absence of a character in the graphic storytelling to see if the effects on narrative transportation differ.

Response 2: We thank the reviewer for these valuable comments. As relevant research on this topic is limited, we have made an effort to identify and discuss related studies in the manuscript (Page 8, Lines 288-290 and Lines 294-299). Additionally, as suggested, future research directions are addressed in the limitation and future orientation section (Page 10, Lines 371-373).

Comments 3: Managerial Implications - Specificity: The managerial implications are generally good but could be more concrete in certain suggestions.The suggestion to optimize advertising by "displaying cooking processes, particularly for semi-finished foods" isa good one. However, the advice for ensuring visuals have "high figure-ground contrast, simple layouts, and visual clarity" is somewhat generic. The authors should provide more specific examples or actionable advice for designers. For instance, rather than just "simple layouts," perhaps suggest "clean, uncluttered layouts with clear visual hierarchy," or "using universally understood icons for preparation steps." Illustrating with concrete examples of successful graphic storytelling in existing food packaging could also strengthen this section.

Response 3: Thank you for the suggestion. The managerial implications section has been revised to include more specific examples and actionable advice for designers (Page 10, Lines 350-357).

Comments 4: Limitations and Future Research - Depth:The limitations are identified, but some could be expanded to suggest more innovative or impactful future research directions.The first limitation mentions static vs. video formats and briefly notes that dynamic formats might have a stronger impact due to sound, motion, and music. The suggestion to investigate "how dynamic narratives influence consumers' decision-making processes" is quite broad. For dynamic narratives, specify what aspects of sound, motion, or music could be investigated (e.g., the role of tempo in conveying preparation speed, specific musical cues for different food types). For the self-reported data limitation, beyond "real-world settings," consider suggesting specific physiological measures (e.g., eye-tracking to confirm attention to graphic elements) or behavioral measures (e.g., actual purchase data in a controlled store environment) to complement self-report. For the "other potential factors" limitation (e.g., food taste and personality), suggest how these could be integrated into the current model, perhaps as moderators or additional mediators, to build a more holistic framework. For example, how might perceived food taste interact with graphic storytelling to influence purchase intention?

Response 4: Thank you for the suggestion. In the limitations section, we have included the following additions: (1) the potential influence of the interaction between background music in video-based storytelling and food attributes on consumer food preferences, (2) the potential to replicate the findings of this study using eye-tracking technology, and (3) the possible moderating effects of individual personality traits. (Page 10, Lines 364-388).

Reviewer 2 Report

Comments and Suggestions for Authors

Dear Authors, I liked the paper; however, it should be improved before publishing.

Please, find my comments below.

Abstract. The relevance of the research is presented. However, the approach and method are not indicated. One can only guess that the research was quantitative (but it was experimental). Also, I would suggest clearly indicating the theories what the research is drawn on. I.e., Cognitive fluency theory and Narrative transportation theory. Clearly naming theories, contexts and methods in the abstracts help the researchers who provide systematic literature reviews and, thus, enhance chances to be cited.

Introduction. Previous exploration of the field and the main findings are not discussed to reveal the gap. The research methodology and main findings are not presented. Also, the contribution to the field of knowledge is not clear.

Literature review. H3 should be revised. Currently it is too complex. I also suggest demonstrating hypotheses on the conceptual model.

Study 1. More explanation is needed why it was chosen to use advertising instead of packaging.

Study 2. Study 2 has absolutely no relation to packaging.

Discussion. The authors should more precisely emphasize their contribution to the field of knowledge. What did this research add to the theories (narrative transportation and cognitive fluency). Currently, the findings are discussed without clear emphasis on the new findings. On the other hand, a contribution to the Transportation-Imagery Model is clearly presented.

Another my concern is that packaging is omitted. In the limitation-devoted section the Authors speak about video-based narratives, sounds, etc. How are these related to packaging? I suggest revising the title of the paper.

Conclusions. The text in the conclusion section looks more like a presentation of the results. It does not conclude anything.

Author Response

Dear Authors, I liked the paper; however, it should be improved before publishing.

Please, find my comments below.

Comments 1: Abstract. The relevance of the research is presented. However, the approach and method are not indicated. One can only guess that the research was quantitative (but it was experimental). Also, I would suggest clearly indicating the theories what the research is drawn on. I.e., Cognitive fluency theory and Narrative transportation theory. Clearly naming theories, contexts and methods in the abstracts help the researchers who provide systematic literature reviews and, thus, enhance chances to be cited.

Response 1: Thank you for the suggestion. The abstract has been revised to include the transportation-imagery model, and we have replaced “studies” with “experimental studies” to enhance readability (Page 1, Lines 17-20).

Comments 2: Introduction. Previous exploration of the field and the main findings are not discussed to reveal the gap. The research methodology and main findings are not presented. Also, the contribution to the field of knowledge is not clear.

Response 2: Thank you for the suggestion. We have rephrased the research gap and added details about the experimental method, research findings, and contributions (Pages 1-2, Lines 39-54).

Comments 3: Literature review. H3 should be revised. Currently it is too complex. I also suggest demonstrating hypotheses on the conceptual model.

Response 3: To improve readability, the sentence has been revised as follows: “We hypothesize that cognitive fluency mediates the relationship between graphic storytelling and narrative transportation (H2a), which subsequently leads to an increased purchase intention (H2b).” (Page 3, Lines 118-120).

Comments 4: Study 1. More explanation is needed why it was chosen to use advertising instead of packaging.

Response 4: We have added an explanation regarding why advertising was chosen instead of packaging (see Page 6, Lines 199-203).

Comments 5: Study 2. Study 2 has absolutely no relation to packaging.

Response 5: Thank you. We have discussed the comparison between study 1 and study 2 in the discussion part (see Page 6, Lines 199-203) and revised the title as “Applying Visual Storytelling in Food Marketing: The Effect of Graphic Storytelling on Narrative Transportation and Purchase Intention” (Page 1, Lines 1-3).

Comments 6: Discussion. The authors should more precisely emphasize their contribution to the field of knowledge. What did this research add to the theories (narrative transportation and cognitive fluency). Currently, the findings are discussed without clear emphasis on the new findings. On the other hand, a contribution to the Transportation-Imagery Model is clearly presented.

Response 6: The contribution has been discussed in the discussion section (see Page 9, Lines 304-306 and Lines 326-329).

Comments 7: Another my concern is that packaging is omitted. In the limitation-devoted section the Authors speak about video-based narratives, sounds, etc. How are these related to packaging? I suggest revising the title of the paper.

Response 7: Thank you. The title has been revised (see Page 1, Lines 1-3).

Comments 8: Conclusions. The text in the conclusion section looks more like a presentation of the results. It does not conclude anything.

Response 8: Thank you. The conclusion has been rewritten (see Page 10, Lines 390-394).

Reviewer 3 Report

Comments and Suggestions for Authors

  • As a research title “Applying Visual Storytelling in Food Packaging: The Effect of Graphic Storytelling on Narrative Transportation and Purchase Intention” I congratulate the authors for their good choice
  • Please describe the employed methods in the abstract ( of the two conducted studies)
  • The first part of the introduction “In today’s competitive marketplace, brands are increasingly turning to engaging visual strategies to capture consumer attention. One such strategy is the use of graphic storytelling on product packaging, particularly in the food industry, where it is increasingly used to illustrate food preparation steps, or to provide cooking instructions, for prepack aged meals” Needs a strong reference
  • This sentence is the core of the research gap , but without reference “Despite its growing presence in the context of food packaging design, the effects of graphic storytelling on food purchase intention, as well as the underlying psychological mechanisms, remain underexplored”, please elaborate more
  • The research motive and contribution need more effors in the introduction section
  • Summarize the research structure at the end of the introduction section
  • The theoretical background is well written in the literature review section
  • Justify the adequacy of sample size in study 1
  • The results of study 1 is interesting
  • Justify the adequacy of sample size in study 2
  • The comparison between study 1 and study 2 nee to be strengthened to demonstrate the significance of your study
  • The conclusion part is poor
  • No section for limitation and future study opportunities

Author Response

As a research title “Applying Visual Storytelling in Food Packaging: The Effect of Graphic Storytelling on Narrative Transportation and Purchase Intention” I congratulate the authors for their good choice

Comments 1: Please describe the employed methods in the abstract (of the two conducted studies)

Response 1: Thank the reviewer. The revision has been made (see Page 1, Lines 17-20).

Comments 2: The first part of the introduction “In today’s competitive marketplace, brands are increasingly turning to engaging visual strategies to capture consumer attention. One such strategy is the use of graphic storytelling on product packaging, particularly in the food industry, where it is increasingly used to illustrate food preparation steps, or to provide cooking instructions, for prepackaged meals” Needs a strong reference

Response 2: We appreciate the reviewer’s suggestion. The relevant references have been added (see Page 1, Lines 32-34).

Comments 3: This sentence is the core of the research gap , but without reference “Despite its growing presence in the context of food packaging design, the effects of graphic storytelling on food purchase intention, as well as the underlying psychological mechanisms, remain underexplored”, please elaborate more

Response 3: Many thanks to the reviewer for the insightful comments. While there are studies on storytelling in food consumption, to the best of our knowledge, no research has specifically investigated the effect of graphic storytelling—using visuals to convey narrative information—on food consumption. We have discussed this in the introduction section to enhance the manuscript's clarity and readability (see Pages 1-2, Lines 39-44).

Comments 4: The research motive and contribution need more efforts in the introduction section

Response 4: We appreciate the reviewer’s valuable suggestions. The motivation and contributions have been discussed in the introduction section (see Page 2, Lines 42-46 and Lines 51-54).

Comments 5: Summarize the research structure at the end of the introduction section

Response 5: The research structure has been summarized at the end of the introduction section (see Page 2, Lines 47-51). Once again, thanks!

Comments 6: The theoretical background is well written in the literature review section

Response 6: Thanks

Comments 7: Justify the adequacy of sample size in study 1

Response 7: Many thanks! The sample size issue has been discussed in the limitation part (see Page 10, Lines 373-378).

Comments 8: The results of study 1 is interesting

Response 8: Thanks!

Comments 9: Justify the adequacy of sample size in study 2

Response 9: Many thanks! As you mentioned above, this issue has been discussed in the limitation part (see Page 10, Lines 373-378).

Comments 10: The comparison between study 1 and study 2 need to be strengthened to demonstrate the significance of your study

Response 10: The comparison between study 1 and study 2 has been discussed in the discussion part (see Page 6, Lines 199-203).

Comments 11: The conclusion part is poor

Response 11: Thank you for the comment. The revision has been made (see Page 10, Lines 390-394).

Comments 12: No section for limitation and future study opportunities

Response 12: The Limitation and Future Directions has been discussed (see Page 10, Lines 360-388).

Round 2

Reviewer 1 Report

Comments and Suggestions for Authors

Review (second version): Applying Visual Storytelling in Food Marketing: The Effect of Graphic Storytelling on Narrative Transportation and Purchase Intention

The authors have made considerable efforts to revise the manuscript, incorporating several valuable suggestions from the previous review round. The paper addresses a highly relevant topic in current food marketing trends, particularly the increasing use of visual strategies to engage consumers. The two experimental studies offer interesting insights into the interplay between graphic storytelling, cognitive fluency, and narrative transportation in influencing purchase intention. However, despite these revisions, there are still areas that could be enhanced to improve the clarity, rigor, and overall impact of the research.

  1. Discussion of Absence of Clearly Identifiable Characters: The authors expanded on the point that graphic storytelling can enhance narrative transportation even without identifiable characters. They offered explanations such as consumers implicitly inferring a character (e.g., a chef or themselves), and the stronger role of verisimilitude and plot coherence. While the expansion is noted, the theoretical backing for narrative transportation in the absence of explicit characters could be strengthened. The argument about "implicit inference" is plausible but relies on speculation without deeper theoretical grounding or empirical evidence specific to visual marketing. The analogy to musical involvement is interesting but might not directly transfer to visual narratives without further elaboration on the shared psychological mechanisms. Although the paper cites research on step-by-step processes enhancing transportation, a clearer connection between this mechanism and the absence of a character could be made. To bolster this section, consider drawing on broader theoretical frameworks: Explore literature from cognitive psychology, semiotics, or visual communication that discusses how viewers construct narratives or engage with visual sequences without explicit human protagonists. Strengthening the "implicit inference" argument: If possible, cite studies that provide more direct evidence or theoretical models for this phenomenon in visual contexts. Elaborating on the role of verisimilitude and plot coherence: Explain in more detail how these elements specifically compensate for the lack of a character in graphic storytelling to induce transportation.
  2. Theoretical Contributions: The paper highlights its extension of the Transportation-Imagery Model by positioning cognitive fluency as an important antecedent of narrative transportation. While the mediation analyses clearly demonstrate the empirical support for cognitive fluency's role, the discussion could further emphasize the specific novelty of this theoretical extension within the context of graphic storytelling. The link between fluency and immersion is present in broader cognitive literature, so clarifying how this paper's findings uniquely contribute to the Transportation-Imagery Model, particularly as applied to visual/graphic narratives in marketing, would enhance its theoretical impact. The authors should articulate more explicitly how their research provides a new theoretical pathway or nuance within the Transportation-Imagery Model, distinguishing its contribution from existing understandings of fluency in other narrative forms (e.g., purely text-based). This would strengthen the argument for the paper's original theoretical contribution.
  3. Clarity of "Graphic Storytelling" Concept: The definition of graphic storytelling as a series of images in a logical sequence is clear. However, briefly discussing the range of what constitutes "graphic storytelling" in marketing (from simple sequential steps to more complex visual narratives like comics) and positioning where this study's manipulation fits could add valuable context and guide future research.
  4. The text (page 6) and Figure 2 indicates that participants received 1 yuan in total. Is this correct?

Overall, the manuscript presents a valuable empirical contribution to the literature on visual storytelling in food marketing. Addressing the above points will further strengthen the paper's theoretical robustness, methodological transparency, and practical implications.

Author Response

Review (second version): Applying Visual Storytelling in Food Marketing: The Effect of Graphic Storytelling on Narrative Transportation and Purchase Intention

The authors have made considerable efforts to revise the manuscript, incorporating several valuable suggestions from the previous review round. The paper addresses a highly relevant topic in current food marketing trends, particularly the increasing use of visual strategies to engage consumers. The two experimental studies offer interesting insights into the interplay between graphic storytelling, cognitive fluency, and narrative transportation in influencing purchase intention. However, despite these revisions, there are still areas that could be enhanced to improve the clarity, rigor, and overall impact of the research.

Comments1: Discussion of Absence of Clearly Identifiable Characters: The authors expanded on the point that graphic storytelling can enhance narrative transportation even without identifiable characters. They offered explanations such as consumers implicitly inferring a character (e.g., a chef or themselves), and the stronger role of verisimilitude and plot coherence. While the expansion is noted, the theoretical backing for narrative transportation in the absence of explicit characters could be strengthened. The argument about "implicit inference" is plausible but relies on speculation without deeper theoretical grounding or empirical evidence specific to visual marketing. The analogy to musical involvement is interesting but might not directly transfer to visual narratives without further elaboration on the shared psychological mechanisms. Although the paper cites research on step-by-step processes enhancing transportation, a clearer connection between this mechanism and the absence of a character could be made. To bolster this section, consider drawing on broader theoretical frameworks: Explore literature from cognitive psychology, semiotics, or visual communication that discusses how viewers construct narratives or engage with visual sequences without explicit human protagonists. Strengthening the "implicit inference" argument: If possible, cite studies that provide more direct evidence or theoretical models for this phenomenon in visual contexts. Elaborating on the role of verisimilitude and plot coherence: Explain in more detail how these elements specifically compensate for the lack of a character in graphic storytelling to induce transportation.

Response1: We thank the reviewer for the insightful and constructive comments, which have greatly helped us improve the manuscript. First, we have elaborated on the shared psychological mechanisms between graphic storytelling and music, specifically focusing on mental simulation (see Page 8, Lines 294–296).

Second, while many studies have examined narrative transportation, to the best of our knowledge, no research has specifically explored the effect of storytelling without characters on narrative transportation in visual contexts. To support our "implicit inference," we have cited relevant research on music as indirect evidence (see Page 8, Lines 292–298). We also acknowledge the need for future research to investigate the potential role of identifiable characters in narrative transportation by manipulating the presence or absence of characters in graphic storytelling. This has been discussed in the limitations and future research section to enhance the manuscript's clarity and contribution (see Page 10, Lines 388–390).

Third, we have expanded our explanation of how verisimilitude and coherence may compensate for the absence of characters, as suggested (see Page 8, Lines 300-303).

Comments2: Theoretical Contributions: The paper highlights its extension of the Transportation-Imagery Model by positioning cognitive fluency as an important antecedent of narrative transportation. While the mediation analyses clearly demonstrate the empirical support for cognitive fluency's role, the discussion could further emphasize the specific novelty of this theoretical extension within the context of graphic storytelling. The link between fluency and immersion is present in broader cognitive literature, so clarifying how this paper's findings uniquely contribute to the Transportation-Imagery Model, particularly as applied to visual/graphic narratives in marketing, would enhance its theoretical impact. The authors should articulate more explicitly how their research provides a new theoretical pathway or nuance within the Transportation-Imagery Model, distinguishing its contribution from existing understandings of fluency in other narrative forms (e.g., purely text-based). This would strengthen the argument for the paper's original theoretical contribution.

Response2: Thank you for this helpful suggestion. We have revised the discussion section to more clearly delineate the theoretical distinction between our findings and prior research focused on text-based storytelling (see Page 9, Lines 320–324). Additionally, we have explicitly emphasized how our findings extend the Transportation-Imagery Model, as suggested (see Page 9, Lines 334–336).

Comments3: Clarity of "Graphic Storytelling" Concept: The definition of graphic storytelling as a series of images in a logical sequence is clear. However, briefly discussing the range of what constitutes "graphic storytelling" in marketing (from simple sequential steps to more complex visual narratives like comics) and positioning where this study's manipulation fits could add valuable context and guide future research.

Response3: We appreciate the reviewer’s suggestion. We have added a brief discussion of the various forms that graphic storytelling can take in marketing and clarify the format adopted in this research (see Page 2, Lines 58-62).

Comments4: The text (page 6) and Figure 2 indicates that participants received 1 yuan in total. Is this correct?

Response4: Thank you for pointing this out. Yes, that is correct—participants received ¥1 RMB (about $0.14 U.S. dollars) in total, as indicated in the text (Page 6, Line 232) and Figure 2.

Overall, the manuscript presents a valuable empirical contribution to the literature on visual storytelling in food marketing. Addressing the above points will further strengthen the paper's theoretical robustness, methodological transparency, and practical implications.

Response: Thanks: )

Reviewer 3 Report

Comments and Suggestions for Authors

accepted

Author Response

Many thanks for your comments: )